# The Involvement of Antioxidants in Cognitive Decline and Neurodegeneration: *Mens Sana in Corpore Sano*

**DOI:** 10.3390/antiox13060701

**Published:** 2024-06-07

**Authors:** Claudio Nazzi, Alessio Avenanti, Simone Battaglia

**Affiliations:** 1Dipartimento di Psicologia, Università degli Studi di Torino, 10134 Torino, Italy; claudio.nazzi@studio.unibo.it; 2Centro Studi e Ricerche in Neuroscienze Cognitive, Dipartimento di Psicologia “Renzo Canestrari”, Alma Mater Studiorum Università di Bologna, Campus di Cesena, 47521 Cesena, Italy; alessio.avenanti@unibo.it; 3Neuropsychology and Cognitive Neuroscience Research Center (CINPSI Neurocog), Universidad Católica del Maule, Talca 3460000, Chile

**Keywords:** memory, cognitive impairment, Alzheimer’s disease, neurodegeneration

## Abstract

With neurodegenerative disorders being on the rise, a great deal of research from multiple fields is being conducted in order to further knowledge and propose novel therapeutic interventions. Among these investigations, research on the role of antioxidants in contrasting cognitive decline is putting forward interesting and promising results. In this review, we aim to collect evidence that focused on the role of a variety of antioxidants and antioxidant-rich foods in improving or stabilizing cognitive functions, memory, and Alzheimer’s disease, the most common neurodegenerative disorder. Specifically, we considered evidence collected on humans, either through longitudinal studies or randomized, placebo-controlled ones, which evaluated cognitive performance, memory abilities, or the progression level of neurodegeneration. Overall, despite a great deal of variety between study protocols, cohorts of participants involved, neuropsychological tests used, and investigated antioxidants, there is a solid trend that suggests that the properties of antioxidants may be helpful in hampering cognitive decline in older people. Thus, the help of future research that will further elucidate the role of antioxidants in neuroprotection will lead to the development of novel interventions that will take into account such findings to provide a more global approach to treating neurodegenerative disorders.

## 1. Aging and Cognitive Decline

The natural process of senescence or aging involves a multifaceted array of changes involving genetic, lifestyle, and environmental factors [1,2]. It also constitutes a predominant risk factor for many chronic diseases, such as cardiovascular diseases, cancer, and neurodegeneration [3]. The increased susceptibility to these conditions is related to a progressive inability to maintain a homeostatic balance, which results in a deterioration of biochemical and physiological functions [4,5]. Moreover, aging is associated with oxidative damage to cells. According to the free radical theory of aging, which has been subsequently renamed the oxidative stress theory of aging, functional losses related to aging are caused by the oxidative damage buildup to macromolecules such as DNA, lipids, and proteins by reactive oxygen and nitrogen species [6,7]. In the case of neurodegenerative disorders, however, additional detrimental mechanisms come into play, such as the buildup of protein aggregates [8]. This process also fosters neuroinflammation, which is not only a byproduct of neurodegeneration but also plays a crucial role in its development as it is present even before protein aggregation [9]. Additionally, the degeneration induced by protein aggregation engenders a detrimental cycle between the exhaustion of phagocytosis and the stimulation of oxidative genes, leading to the overproduction of toxic oxidative species [10]. Furthermore, oxidative stress may be increased by exogenous sources, for example, pesticides [11,12], which are known to be one of the causes of the pathogenesis of neurodegenerative disorders [13,14].

However, recent evidence suggests that antioxidants may be critical in contrasting the development of a series of neurodegenerative diseases [15,16,17] (Figure 1). Generally, neurodegeneration may arise following the insurgence of mild cognitive impairment (MCI). This condition was first defined by Petersen and colleagues at the end of the past century [18] and refers to an impairment in cognition that is above the norm but does not have repercussions for the ability to carry out daily activities. MCI can generally be further distinguished as amnestic and non-amnestic. In the first case, there is a decline in the ability to learn new information or recall stored information, while in the latter, memory functionality is still intact but another cognitive domain is negatively impacted (social functioning, language, visuospatial function, complex attention, or executive functioning) [19]. The prevalence of amnestic MCI is superior to its other manifestations [20], suggesting that memory may be the most impacted domain. Accordingly, memory impairments represent one of the most commonly reported complaints by aging people [21,22]. As a matter of fact, alongside neurotransmitters [22,23,24,25], reactive oxygen species are an important part of efficient memory function in physiological conditions, but during aging, the increasing number of reactive oxygen species overwhelms the system, leading to damage [26]. In this context, lower blood concentration levels of L-ergothioneine, a dietary antioxidant [27], have been found in MCI patients compared to age-matched controls [28]. Accordingly, daily consumption of ergothioneine supplements has proven to foster improvements in both memory and attention in a sample of participants that included patients with MCI [29].

## 2. The Gut–Brain Axis

The gut–brain axis (GBA) is a complex, bidirectional communication system involving the central nervous system, the enteric nervous system, and the gut microbiota. This axis regulates key physiological processes, including food intake, immune function, and mood. The GBA allows gut-derived molecules and microbe-derived neuroactive compounds to influence brain function, and vice versa. The vagus nerve, a crucial component of the autonomic nervous system, acts as a neurometabolic sensor, detecting gut microbiota metabolites and transmitting this information to the brain [30].

Converging and recent evidence is suggestive of the central role of the GBA as a crucial substrate involved in the process of neurodegeneration [31]. Specifically, gut microbiota composition perturbations lead to increased oxidative stress, neuronal death, and neuroinflammation, culminating in the development of disorders such as Alzheimer’s disease [32,33] and Parkinson’s disease [34,35]. Disturbances in gut microbiota—characterized by a reduction in beneficial species and an increase in pathogenic, pro-inflammatory bacteria—lead to the production of toxic metabolites and pro-inflammatory cytokines. These changes compromise the integrity of the gut epithelial barrier, activate local and distant immune responses, and cause dysregulation of enteric neurons and glia. Subsequent dysfunctions in the blood–brain barrier emerge, triggering neuroinflammatory responses and predisposing neurons and glial cells to apoptosis, particularly in brain regions such as the hippocampus and cerebral cortex, contributing to the development of neurodegenerative disorders [36].

The consumption of antioxidants fosters neuroprotective effects and can modulate gut microbiota composition, increasing beneficial bacteria and reducing pathogenic species, also improving gut barrier integrity and blood–brain barrier permeability [31]. Crucially, a series of antioxidants have been found in lower concentrations in frail and cognitively impaired individuals compared to healthy controls [37]. Accordingly, the evidence suggests that antioxidant administration through a series of nutritional supplements could help contrast cognitive decline and reduce the chances of the development of more serious conditions, as will be discussed in the following paragraphs.

## 3. You Are What You Eat: Neuroprotective Effects of Antioxidants

### 3.1. Healthy Population and Normal Aging

A breadth of studies on the effects of antioxidants on cognition have been conducted on various cohorts of participants. A series of these have been carried out in a longitudinal setting, involving a large number of participants evaluated over a large time window, providing stable correlational evidence (Table 1). As an example, Wengreen and colleagues carried out a longitudinal study including elderly participants, in which dietary habits and cognitive performance were evaluated at three follow-up sessions over the course of 7 years [38]. Neuropsychological profiles were assessed with the Modified Mini-Mental State Examination (3MS) [39], which includes additional tests compared to the original version, designed to sample a wider range of functions. The results indicate that higher vitamin C and E intakes may help preserve cognitive function in the elderly, as evidenced by the fact that participants with the highest levels of vitamin intake had better cognitive performance at baseline and maintained a higher 3MS score at follow-up interviews.

Another longitudinal study [40] involving participants 45 years of age or older assessed cognitive performance at baseline and 5 years later, while also evaluating antioxidant intake by means of a questionnaire. Neuropsychological tests included assessments of memory, speed of cognitive processes, and cognitive flexibility. Among the different antioxidants that were part of the evaluation, lignans revealed the most interesting results. A higher intake of these compounds was associated with lower cognitive decline, and people in the lowest quintile of lignan intake showed a 3.5 times faster decline rate in global cognitive function. Additionally, people in the lowest quintile for vitamin E intake showed a greater decline, specifically in memory performance. The results for the other antioxidants analyzed did not reveal significant associations with cognitive decline, and the authors suggest this may be due to the fact that the sample included middle-aged participants, which do not show a link between cognitive decline and antioxidant intake [46]. In any case, the results suggest that lignans may be beneficial as a protective factor against cognitive decline.

Total antioxidant capacity (TAC) is a measure that can assess the antioxidant status of biological samples [47]. Peng and colleagues [41] aimed to study the eventual association between this measure and cognitive performance in a large sample of older adults who took part in the National Health and Nutrition Examination Survey, which aims to assess the nutritional status of the United States population [48]. The test used to evaluate the cognitive profile included assessments of memory, fluency, and executive function. The results showed that participants with higher levels of TAC had a lower risk of impaired cognitive function compared to people with lower TAC, suggesting that antioxidant action may have a protective effect on cognition. Importantly, this study corroborates older findings, which suggested that vitamin E intake is correlated with better cognitive functioning in older people [49,50]. Potential protective effects on brain functioning have also been investigated in relation to flavonoids. These compounds are a group of natural substances primarily found in plants and hold favorable antioxidant properties [51]. A recent review conducted by Ramezani and colleagues [52] collected evidence from preclinical and clinical studies focusing on the role of flavonoids in cognitive decline. The authors suggest promising evidence concerning the role of flavonoids as neuroprotective compounds and suggest that the main mechanisms upon which they act are twofold, namely neurotransmission regulation and neurogenesis, synaptic plasticity, and neuronal survival.

Other than evidence on general cognitive function, some investigations also provided proof of improvements specifically for memory performance. For instance, Perrig and colleagues [42] recruited volunteers 65 years of age and older and tested their plasma vitamin levels as well as different subcomponents of memory. Their results showed that levels of vitamin C and β-carotene were positively correlated with free recall, recognition, and semantic memory, suggesting a protective role of these compounds on memory function. In a similar study, Perkins and colleagues [43] evaluated memory performance in older adults who took part in the third iteration of the National Health and Nutrition Examination Survey [48]. Memory performance was assessed with both word and story recall in order to evaluate long-term memory. The results highlight that low levels of vitamin E were linked to worse memory performance, even after adjusting for confounding factors, thus suggesting its protective effect on memory deterioration.

The influence of antioxidants has also been studied in a younger sample of volunteers, specifically those with weight problems. Cannavale and colleagues [44] studied antioxidant serum concentration levels and dietary reports in overweight participants, while also assessing visual memory with a spatial reconstruction task. The results revealed that higher serum concentrations of lutein, a carotenoid with a strong antioxidant effect [53], are associated with greater relational memory performance. Accordingly, the authors suggest that one of the possible mechanisms at play in this context is the modulation of the oxidative stress pathways carried out by lutein, which has been observed in relation to eye health but may be at play similarly even at the neural level [54].

Other than the investigations outlined above, which brought forward correlational evidence, randomized, placebo-controlled studies reveal promising results, corroborating these findings and providing causal evidence by directly manipulating dietary regimes or by providing daily doses of supplements and evaluating their impact on cognitive functioning (Table 2). Evans and colleagues [55] conducted a study on post-menopausal women administering a daily dose of resveratrol or a placebo for two weeks. Their results highlight a significant improvement in cognitive performance only in the experimental group, which also correlated with cerebrovascular responsiveness. Similarly, Martínez-Lapiscina and colleagues [56] evaluated the impact of the Mediterranean diet supplemented with either extra-virgin olive oil or mixed nuts compared to a different, low-fat control diet in a sample of older adults. Participants were instructed to follow their assigned diet and were tested after a mean of 6.5 years since the beginning. The tests included the Mini-Mental State Examination (MMSE) [57], which assesses orientation to time and place, registration, attention and calculation, recall, language, and visual construction, as well as the Clock Drawing Test [58], which requires language comprehension, visuospatial abilities, working memory, attention, and abstract thinking for successful completion. The results show that both groups that followed the enriched Mediterranean diets showed significantly higher global cognitive scores compared to the control group. Since both extra-virgin olive oil [59] and nuts [60] are known to contain a variety of antioxidants, the authors attribute better cognitive profiles to the properties of these two foods. Notwithstanding, the lack of baseline measurements of cognitive abilities prevented the authors from defining the presence of changes over time. Thus, Valls-Pedret and colleagues [61] subsequently conducted a similar study with the specific aim of addressing this issue. They followed the same experimental procedure as above and additionally administered neuropsychological tests both at the beginning and the end of the trial. The tests included the MMSE as well as the Rey Auditory Verbal Learning Test [62] and a subtest of the Wechsler Memory Scale [63], both aimed at evaluating verbal memory performance. The results from this study highlight an improvement in cognition for participants in the so-called ‘Mediterranean diet’ groups and cognitive decline in participants following the control diet, supporting and complementing findings from the previous study. Much like their colleagues, Valls-Pedret and associates suggest that the improvements can be attributed to the antioxidant properties of extra-virgin olive oil and nuts.

One such example is an experiment conducted by Bookheimer and colleagues [64], in which elderly participants were instructed to consume a daily dose of pomegranate juice, which, compared to that of other fruits, yields the highest amount of antioxidants [65], or a control drink and were tested for memory performance both at the beginning and the end of the investigation. The tests used for memory assessment were the Buschke–Fuld Selective Reminding Task [66], which assesses short and long-term memory, and an experimental unrelated word pair association task [67]. Other than behavioral data, participants also underwent functional magnetic resonance imaging (fMRI) to test for differences in brain activations. The results highlighted that the experimental group yielded a significantly better memory performance at the post-treatment assessment compared to the control group. The fMRI results support these findings by showing that the experimental group showed the recruitment of additional brain areas in the post-treatment session compared to the baseline, while no differences were detected in the control group. Thus, the authors suggest that pomegranate juice may increase task-related brain activations and consequently improve memory abilities in older adults. In summary, these studies have shown that higher levels of vitamin C and vitamin E intake are associated with better cognitive performance [38,42], and the same is true for lignans [40]. Moreover, participants with the lowest vitamin E intake were shown to be subject to faster cognitive decline compared to participants with the highest intake levels [40], as well as worse memory performance [43]. Additionally, some evidence suggests an important role of carotenoids as well, as highlighted by better verbal [42] and visual [44] memory performance. Despite these specific compounds, the overall amount of antioxidants is also of crucial importance, since low TAC levels are associated with more pronounced cognitive decline [41].

These findings have also been corroborated by randomized, placebo-controlled investigations. A Mediterranean diet supplemented with either extra-virgin olive oil or mixed nuts revealed a positive effect on cognition, as participants assigned to these groups had better overall cognitive performance scores compared to people following a low-fat control diet [56], while also showing cognitive improvements over time while participants in the control diet showed cognitive decline [61]. Similarly, daily consumption of pomegranate juice leads to better cognitive performance compared to a placebo substance [64], much like resveratrol [55].

### 3.2. Neurodegeneration

Dementia can be defined as any impairment that causes a decisive cognitive decline compared to the previous level, which impacts negatively on everyday activities—this evolving process can be encapsulated in the concept of neurodegeneration. It is a syndrome that can be determined by a series of diseases, mainly neurodegenerative ones [68]. Among these, the most common is Alzheimer’s disease (AD), with an estimated 6.7 million Americans suffering from this condition as of 2023 [69]. It is most commonly characterized by progressive amnestic deterioration featuring neurofibrillary tangles at the level of temporal lobe structures, but in some cases can be defined by impairments in other behavioral domains, such as emotion, social interaction, arousal and regulation, sensorimotor function, and pain sensation [70,71,72,73,74]. Given that estimates suggest that its prevalence will dramatically increase with time [75], research is focusing on this issue with increasing interest. Accordingly, a series of studies have focused on the role of antioxidants as a preventive measure to contrast the development of this neurodegenerative disease (Table 3). One of the first examples comes from Engelhart and colleagues [45], who carried out a longitudinal investigation based on the Rotterdam Study, a cohort study aimed at evaluating the presence of a series of diseases in older people [76]. Participants underwent a cognitive assessment comprising the MMSE and the Geriatric Mental State schedule, an interview aimed at classifying patients based on their symptom profile and track changes over time [77]. Additionally, dietary habits were investigated in order to provide an estimate of antioxidant consumption. After a mean 6-year follow-up visit, the neuropsychological tests were readministered in order to assess cognitive decline. The results highlighted that a higher intake of both vitamin C and vitamin E was associated with a lower risk of developing AD, providing insights into the role of these two vitamins in neurodegeneration prevention, even though these results do not provide causal evidence.

More robust data comes from randomized, placebo-controlled trials. One such example is the study by Akhondzadeh and colleagues [78], which aimed at evaluating the effectiveness of saffron, which yields potent antioxidant properties, thanks to the presence of crocin [83], in the treatment of mild to moderate AD. To do so, they enrolled probable AD patients in a 16-week study to evaluate cognitive changes assessed with the AD assessment scale-cognitive subscale [84], a tool that combines a series of tasks to investigate the level of cognitive dysfunction in AD patients. Participants were split into two groups, and those in the experimental group were given a daily dose of saffron while the controls were administered a placebo. The results showed that, after 16 weeks, the experimental group was characterized by significantly lower levels of cognitive impairment compared to the control group, suggesting the efficacy of saffron on cognitive functions in AD patients. Similarly, Nakagawa and colleagues [79] recruited AD patients for an experiment aimed at evaluating the effect of quercetin, an antioxidant flavonoid [85], on cognition. Participants received a daily dose of either Quergold or Mashiro onions, which contain the highest and lowest levels of quercetin among onion varieties, respectively. Cognitive performance was assessed with the MMSE and the Revised Hasegawa Dementia Scale (HDS-R) [86], a questionnaire that examines the main areas of cognitive functions. The results showed that the consumption of Quergold onions was associated with an improvement in memory recall compared to participants in the control group, suggesting the neuroprotective effects of quercetin in AD patients.

Moreover, some evidence suggests that EGb 761^®^, a *Ginkgo biloba* extract taken from its leaves with potent antioxidant activity [87], can help in hampering cognitive decline in patients suffering from neurodegeneration. Napryeyenko and Borzenko [80] conducted a study on patients suffering from AD or vascular dementia, a type of degeneration primarily caused by cerebrovascular damage [88]. Participants received a dose of EGb 761^®^ or a control substance for 22 weeks and were evaluated before and after the intervention with the Syndrom-Kurz-test (SKT) battery used in the assessment of treatment responses for dementia patients [89]. At the end of the trial, participants in the experimental group showed an improvement in SKT scores, whereas control patients had worse scores compared to the baseline. Furthermore, patients in the experimental group also had better scores in the neuropsychiatric inventory, which assesses neuropsychiatric symptoms in neurodegeneration [90], and in the activities of daily living scale, a questionnaire that evaluates the level of assistance needed by the patient for everyday activities. Hence, the authors suggest that EGb 761^®^ has beneficial effects for patients suffering from neurodegeneration, and not only on a cognitive level but also for their quality of life.

Similar results have been replicated in a subsequent study, corroborating this evidence. Herrschaft and colleagues [81] conducted a similar trial by recruiting patients with AD or vascular dementia and neuropsychiatric symptoms. Participants were split into two groups and given a daily dose of either EGb 761^®^ or a placebo substance for 24 weeks. The results at the end of the trial showed, similarly to the previous study, an improvement in SKT scores for the experimental group only, which was also accompanied by an improvement in neuropsychiatric symptoms measured with the neuropsychiatric inventory. Thus, these results bring additional evidence that the antioxidant properties of *Ginkgo biloba* can have beneficial effects on dementia.

Aside from studies that specifically focused on AD, a study by Choi and colleagues [82] explored the impact of *Spirulina maxima*, a microalga used as a dietary supplement for its abundance of antioxidants [91], on memory in a sample of older adults with MCI. Participants were randomly assigned to the experimental or control group and had to take a daily dose of Spirulina extract or a placebo for 12 weeks, respectively. At baseline and at the end of the intervention, the experimenters administered the Korean version of the Montreal Cognitive Assessment [92], a cognitive screening test able to detect MCI, and a Computerized Neurocognitive Test [93] to each participant. The results revealed that the experimental group showed improvements in visual learning and visual working memory, as well as in vocabulary, while such improvements were not present for the control group. Thus, the authors conclude that *Spirulina maxima* extracts can be effective in contrasting memory decline, providing a safe supplement to prevent degeneration.

In conclusion, the studies described here suggest that higher dietary intakes of some antioxidants can have a positive effect on cognition in neurodegeneration (Table 4). Accordingly, both vitamin C and vitamin E are associated with a reduced risk of progressing towards AD [45]. Moreover, a substantial range of supplements leads to improvements over placebos. As a matter of fact, both saffron [78] and *Ginkgo biloba* [80,81] extracts lead to improvements in cognitive performance, as well as enhancements to the outcomes of secondary quality of life measures. Moreover, higher levels of quercetin administered through food sources that are rich in this substance lead to improvements in memory performance compared to low-quercetin foods [79]. Finally, encouraging data suggests that improvements can be obtained even in the prodromal phases of neurodegenerative diseases, as evidenced by improvements in MCI patients by means of *Spirulina maxima* extract administration [82].

### 3.3. Contrasting Evidence

While the abovementioned investigations provide promising evidence concerning the role of antioxidants as preventive measures to contrast cognitive decline as well as slowing down neurodegeneration, other studies highlighted that antioxidant supplementation did not yield positive effects.

Broman-Fulks and colleagues [99] examined the effects of quercetin supplementation on cognitive performance in healthy adults of various age ranges. Participants were randomized to receive either quercetin or a placebo for 12 weeks. Cognitive performance was assessed using a battery of neuropsychological tests named Central Nervous System Vital Signs [100], which includes measures of memory, executive function, and processing speed. The results indicated no significant improvements in cognitive performance attributable to quercetin supplementation compared to a placebo, regardless of age group. Additionally, some evidence supports the idea that resveratrol may not have positive effects on cognition. As an example, Kennedy and colleagues [101] and Wightman and colleagues [102] assessed the effects of this antioxidant on cognitive performance and cerebral blood flow. Specifically, they assessed cognitive performance with the Rapid Visual Information Processing Task [103] and the Serial Subtraction Task [104] while undergoing near-infrared spectroscopy (NIRS)—a non-invasive brain imaging technique—to evaluate hemodynamic responses. The results of both studies revealed that, while resveratrol or its combination with piperine—a compound with antioxidant properties found in black pepper [105]—had a dose-dependent effect on cerebral blood flow, and this was not reflected in an improvement in cognitive performance.

Some important, additional insights emerge from studies specifically focused on dementia. A study conducted by Kryscio and colleagues [106] suggested that administration of vitamin E, selenium, or a combination of the two, did not affect the chances of developing dementia compared to a placebo. Furthermore, the supplementation of vitamin E in AD patients results in either beneficial or detrimental effects on cognition based on its activity. When vitamin E fosters an antioxidant effect—as measured with oxidized glutathione levels in the blood—cognitive test scores remain stable compared to a placebo condition. However, in patients who did not show oxidative stress prevention, test scores were lower than patients in the placebo condition [107]. While the results of this study support the idea that an antioxidant action can be beneficial in preventing cognitive decline, they warn against the risks of indiscriminate use of supplements, which can lead to opposite outcomes.

## 4. Discussion

Neurodegeneration, the progressive loss of structure and function of neurons [108], represents a formidable challenge in modern neuroscience and health science [109,110]. With an aging global population, the prevalence of neurodegenerative diseases continues to rise, exacting a significant toll on individuals, families, and healthcare systems worldwide [69,75]. Thus, a call for new knowledge is required in order to develop appropriate novel interventions through preclinical, clinical, and translational research [111,112,113]. In this context, the study of antioxidants as protectors of cognitive functions and neurodegeneration is providing uplifting evidence. Furthermore, while healthy eating habits are associated with the prevention of diseases that typically affect older people, this category often consumes inappropriate diets [114]. Moreover, the US dietary guidelines suggest diets that are not fully equipped to satisfy the needs of the aging population compared to other dietary regimes [115]. This calls for the need for further investigations for more tailored suggestions.

The regular activity of neurons requires considerable oxygen consumption, which leads to the production of ROS. The concentration of ROS is regulated by an intricate antioxidant system, which ensures regular functioning. Accordingly, ROS are a key component of memory function, but an excessive number of them, as seen during aging, can lead to neuronal damage and degeneration [26]. ROS can then become harmful molecules known for oxidizing crucial components like enzymes and cytoskeletal proteins, thus posing neurotoxic effects. As organisms age, ROS production rises while antioxidant defenses weaken, leading to heightened neuronal oxidative stress [116,117]. Importantly, ROS are also an important factor in determining long-term potentiation [118], a persistent change in synaptic strength determined by specific patterns of synaptic activity [119]. Accordingly, this phenomenon is defined as *neuroplasticity* and involves the ability of the brain to adapt in response to intrinsic or extrinsic stimuli by reorganizing itself on the structural, functional, and connectional levels [120,121,122]. However, evidence suggests that the aging brain is subject to impairments to plastic abilities [123,124], which could be due to the fact that during aging ROS increase in number and become imbalanced compared to antioxidant levels, which could lead to an impairment in long term potentiation efficiency [125]. For this reason, an adequate amount of antioxidants can help stave off the detrimental effects of oxidative stress on the aging brain, thus alleviating the deficits that can emerge during senescence.

Accordingly, numerous studies have shown a positive effect of antioxidants on brain health. In this review, we collected the evidence gathered throughout the years on both healthy aging people and patients afflicted with neurodegenerative diseases.

Notwithstanding, the investigations carried out to this day also present some limitations. For instance, while the results are fairly consistent, the study designs are substantially different from each other. While some researchers focused on a large number of participants in the span of several years, others only investigated differences over shorter time spans in a small sample [79]. Thus, some differences between dietary styles may be effective in the short term but not over considerably longer periods of time, as these studies did not assess the stability of cognitive improvements with follow-up visits. Furthermore, the assessment of antioxidant intake or manipulation of their intake varied considerably between experiments. In some of the longitudinal studies [38,40,41,42,43,45], antioxidant consumption was purely estimated based on a questionnaire that assessed the dietary habits of participants, which has a chance to not reflect the actual habits of participants. In placebo-controlled studies, antioxidant levels were controlled either with the administration of natural extracts [55,80,81,82] or with foods rich in these compounds [56,61,64,78,79]. In the latter case, precisely controlling the amount of antioxidants is more challenging, which could give rise to additional, non-controllable variability in the data. Finally, antioxidants that reveal promising effects in the healthy population may not be effective in patients with neurodegeneration and vice versa, urging caution when trying to generalize data to other population categories.

Currently, neurodegenerative diseases lack effective treatments, as the approved drug therapies primarily focus on symptom management and slowing disease progression [126,127]. Several factors contribute to the failure of existing treatments, including challenges in understanding the disease’s pathogenesis, the development of drugs targeting specific therapeutic goals, and the subsequent design of clinical trials. However, as highlighted here, natural products offer promising options for alleviating symptoms and slowing disease advancement. The resurgence of natural products as a source of innovative medicines is attributed to the recognition of antioxidants as viable alternatives. These products present a novel therapeutic approach to the treatment of various neurodegenerative conditions, offering considerable therapeutic potential with minimal side effects [128].

## 5. Conclusions and Future Perspectives

In conclusion, there is solid evidence that the effects of antioxidants may help contrast cognitive degeneration in both the healthy population and patients who suffer from neurodegeneration. This knowledge is of critical importance when considering the implications for future research and the clinical interventions that can stem from it as a result. As only one of the studies analyzed herein investigated the underpinnings of the influence of antioxidants on brain function by means of fMRI [64], future investigations will need to integrate psychophysiological methods to further delve into the specific action of these compounds to elucidate the mechanisms that lead to cognitive improvements. Data from healthy samples hints that there may be concrete options for preventive measures aimed at contrasting the future insurgence of cognitive impairments. Accordingly, this evidence could pave the way for a global clinical approach towards cognitive decline in the aging population, suggesting that interventions that integrate neuropsychological methods with nutritional science could represent a fruitful avenue for new treatment options for neurodegeneration [129,130]. A promising yet relatively new avenue is the study of the gut–brain axis, a communication pathway between the nervous and enteric systems that regulates not only gastrointestinal homeostasis but is likely also involved in motivation, affect, and higher cognitive states [30,131]. Interdisciplinary approaches to the study of the interaction between the brain and the body [132,133,134,135] may reveal more insightful ramifications for novel therapies compared to more traditional investigations.

The application of these new hypothetical treatment avenues may help restructure the mental healthcare system by providing an alleviation of the burden on the societal and economic level usually associated with neurodegenerative disorders [136,137,138,139].

## Figures and Tables

**Figure 1 antioxidants-13-00701-f001:**
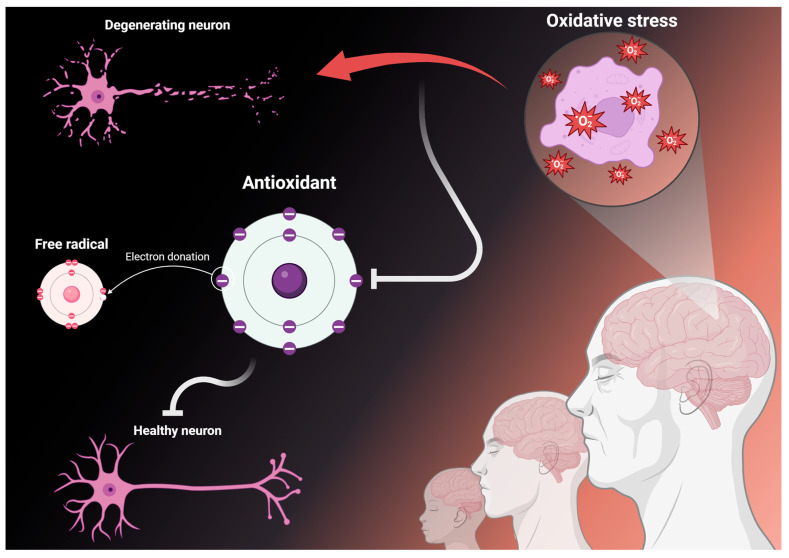
The protective role of antioxidants in neurodegeneration. With aging, a greater imbalance between reactive oxygen species and antioxidants can lead to an increased state of oxidative stress, which can cause damage to cells and, ultimately, lead to neurodegeneration and thus pathological cognitive decline. In this context, a higher concentration of antioxidants, either through the diet or via supplements, can stave off this imbalance and foster a neuroprotective effect by slowing down degeneration, as seen by improvements or stabilization of cognitive performance. Created with BioRender.com.

**Table 1 antioxidants-13-00701-t001:** Summary of correlational studies in healthy participants.

Study	Antioxidant Level Assessment	Participants	Study Duration	Results	Tests Employed
Wengreen et al. [38]	Dietary questionnaire	Healthy, 65 years old or above	7.2-year average follow-up	Higher vitamin C and E intakes associated with preserved cognitive functioning	Modified Mini-Mental State Examination
Nooyens et al. [40]	Dietary questionnaire	Healthy, 45 years old or above	5-year follow-up	Higher lignan intake associated with slower cognitive decline; lower vitamin E intake associated with faster memory performance decline	15-Word Learning Test, Stroop Test, Word Fluency test Letter Digit Substitution Test
Peng et al. [41]	Total antioxidant capacity	Healthy, 60 years old or above	Immediate	Higher levels of total antioxidant capacity associated with lower risk of impaired cognitive function	Immediate Recall Test, Delayed Recall Test, Animal Fluency Test, Digit Symbol Substitution Test.
Perrig et al. [42]	Plasma antioxidant levels	Healthy, 65 years old or above	22-year follow-up	Positive correlation between vitamin C and β-carotene with free recall, recognition, and semantic memory	Computerunterstutzer Gedachtnis-Funktions-Test, Vocabulary (Wechsler Adult Intelligence Scale)
Perkins et al. [43]	Plasma antioxidant levels	Healthy, 60 years old or above	Immediate	Lower levels of vitamin E associated with poor memory	Delayed word recall (Mini-Mental State Examination), Delayed story recall (East Boston Memory Test)
Cannavale et al. [44]	Dietary questionnaire; serum antioxidant levels	Healthy, 25 to 45 years old	Immediate	Higher serum lutein is associated with better relational memory performance	Computerized spatial reconstruction task
Engelhart et al. [45]	Dietary questionnaire	Healthy, 55 years old or above	6-year average follow-up	Higher vitamin C and E intakes associated with lower risk developing Alzheimer’s disease	Mini-Mental State Examination, Geriatric Mental State Schedule

**Table 2 antioxidants-13-00701-t002:** Placebo-controlled studies in non-dementia samples.

Study	Antioxidant/Food	Participants	Study Duration	Results	Tests Employed
Evans et al. [55]	Resveratrol	Post-menopausal women, 45 to 85 years old	14 weeks	Improvements in overall cognitive performance and verbal memory	Rey Auditory Verbal Learning Test, Cambridge Semantic Memory Battery, Double-Span Task, Trail-making Task
Martínez-Lapiscina et al. [56]	Mediterranean diet, extra-virgin olive oil, or mixed nuts supplementation	High vascular risk, 55 to 80 years old	6.5-year average follow-up	Better cognitive performance for participants in both experimental groups compared to a control diet	Mini-Mental State Examination, Clock Drawing Test
Valls-Pedret et al. [61]	Mediterranean diet, extra-virgin olive oil, or mixed nuts supplementation	High vascular risk, 55 to 80 years old	4.1-year median follow-up	Cognitive improvements for participants following both Mediterranean diets; cognitive decline for participants following a control diet	Mini-Mental State Examination, Rey Auditory Verbal Learning Test, Verbal Paired Associates Test (Wechsler Memory Scale), Animal Fluency Test, Color Trails Test, Digit Span (Wechsler Adult Intelligence Scale)
Bookheimer et al. [64]	Pomegranate juice	Self-reported memory complaints, 63.1 and 62 mean ages in years for the experimental and control groups, respectively	4 weeks	Better memory performance for the experimental group compared to the control group; recruitment of additional brain areas during memory tasks for the experimental group	Buschke–Fuld Selective Reminding Task, Verbal Paired Associates Test (Wechsler Memory Scale)

**Table 3 antioxidants-13-00701-t003:** Placebo-controlled studies in patients with neurodegenerative disorders.

Study	Antioxidant/Food	Participants	Study Duration	Results	Tests Employed
Akhondzadeh et al. [78]	Saffron	Probable Alzheimer’s disease, 55 years old or above	16 weeks	Lower levels of cognitive impairment for the experimental group compared to the control group	Mini-Mental State Examination, Cognitive subscale (AD Assessment Scale), Clinical Dementia Rating Scale
Nakagawa et al. [79]	Quercetin (Quergold onions)	Alzheimer’s disease, mean age of 79 years	4 weeks	Improvements in memory recall for the experimental group compared to the control group	Mini-Mental State Examination, Revised Hasegawa Dementia Scale
Napryeyenko and Borzenko [80]	*Ginkgo biloba* special extract EGb 761^®^	Alzheimer’s disease or vascular dementia, 50 years old or above	22 weeks	Improvements compared to baseline in all test scores for the experimental group	Syndrom-Kurz-test, Neuropsychiatric inventory, Activities of Daily Living Scale
Herrschaft et al. [81]	*Ginkgo biloba* special extract EGb 761^®^	Alzheimer’s disease or vascular dementia, 50 years old or above	24 weeks	Improvements compared to baseline in all test scores for the experimental group	Syndrom-Kurz-test, Neuropsychiatric inventory
Choi et al. [82]	*Spirulina maxima* extract	Mild cognitive impairment, mean age of 68 years	12 weeks	Improvements compared to baseline in visual learning, visual working memory, and vocabulary for the experimental group	Montreal Cognitive Assessment, Computerized Neurocognitive Test

**Table 4 antioxidants-13-00701-t004:** Antioxidants that revealed promising effects on cognitive function and their highest-content food sources.

Antioxidant	Food Source
Vitamin C *	Acerola, rose hips, guavas, peppers, peaches [94]
Vitamin E	Sunflower seeds, almonds, hazelnuts, pine nuts, conch [94]
Lignans	Flax seeds, sesame seeds, broccoli, cashew nuts, Brussel sprouts [95]
β-carotene	Sweet potato, carrot, pumpkin, spinach, collards [94]
Lutein	Spinach, kale, parsley, romaine lettuce, pistachio nuts [96]
Quercetin	Dill, fennel leaves, onions, oregano, chili pepper [97]
Resveratrol	Muscadine grape, lingonberry, cranberry, redcurrant, bilberry [98]

* The table reports the five food sources richest in the corresponding antioxidants, excluding concentrated or reinforced products.

## Data Availability

Not applicable.

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
