# Peer review of "The Involvement of Antioxidants in Cognitive Decline and Neurodegeneration: Mens Sana in Corpore Sano"

_antioxidants, 2024, doi:10.3390/antiox13060701_

Round 1

Reviewer 1 Report

The manuscript reviews published  studies focused on the role of antioxidants (molecules, antioxidant rich food, supplements, ...) in improving or or stabilizating symptons of the most common neurodegenerative disorders.

From this deep review authors conclude the effects of antioxidants can be used to prenent cognitive degeneration in healthy people and to contrast patients that already suffer neurodegeneration.

Since there are not effective treatments for neurodegenerative diseases, antioxidant natural products offer a challenge in alternative medicine as a therapeutic approach.

The review is well organized and written, nevertheless, a the manuscript would be much improved including a table containing the description of all this information:

- the name of the antioxidant/molecule/food

- Type of subjects (healthy or pathology) in wich antioxidants were tested

- Time of treatment

- Results

- Performed test 

- References 

Moreover, authors have not mentioned or described any mechanism involved in aging or neurodegeneration, thus, the word "mechanisms" in the title is not appropriate.

- Page 2, line 82, "higher 3MS score" abbreviation has to be described.

- page 5, line 241, SKT should be described as "Short Cognitive Performance Test".

Author Response

Comment 1:

The manuscript reviews published  studies focused on the role of antioxidants (molecules, antioxidant rich food, supplements, ...) in improving or or stabilizating symptons of the most common neurodegenerative disorders.

From this deep review authors conclude the effects of antioxidants can be used to prenent cognitive degeneration in healthy people and to contrast patients that already suffer neurodegeneration.

Since there are not effective treatments for neurodegenerative diseases, antioxidant natural products offer a challenge in alternative medicine as a therapeutic approach.

The review is well organized and written,

Response 1:

We thank the Reviewer for their positive and constructive feedback.

Comment 2:

nevertheless, a the manuscript would be much improved including a table containing the description of all this information:

- the name of the antioxidant/molecule/food

- Type of subjects (healthy or pathology) in wich antioxidants were tested

- Time of treatment

- Results

- Performed test 

- References 

Response 2:

We agree with this recommendation, and accordingly updated the manuscript to include a new table with this information.

Comment 3:

Moreover, authors have not mentioned or described any mechanism involved in aging or neurodegeneration, thus, the word "mechanisms" in the title is not appropriate.

Response 3:

Thanks for this suggestion, we updated the title accordingly.

Comment 4:

Page 2, line 82, "higher 3MS score" abbreviation has to be described.

Response 4:

Thanks, we specified the acronym at first occurrence, which stands for Mini-Mental State Examination.

Comment 5:

page 5, line 241, SKT should be described as "Short Cognitive Performance Test".

Response 5:

Thanks for the suggestion, we specified the acronym as “Syndrom Kurztest” as named by the original authors.

Reviewer 2 Report

I felt that the authors and others collected a little too much good evaluation literature, but they are well studied and could have a positive impact on this field. However, I would like to point out that there are some problems with the current situation.

I think it is well researched and informative, but the duplicate sections are dreadful.

I suspect that in many cases you do not add discussion sessions to your review. At least avoid duplication carefully.

It is good to point out the benefits of antioxidants, but it would be better to include the negative information as well.

More figures could be added.

There is not much visual clarity about this review, which is a drawback.

Regarding the mechanism of oxidative stress generation,in the case of neurodegenerative diseases,protein aggregates and inflammatory mechanisms are involved,which may differ from simple aging. They should probably touch this point a little more. Could this argument support the difference between aging and dementia as described by the author?

 It is interesting,but further explanation of the details of each antioxidant would be useful. What foods contain them, etc., would be of interest. However, several antioxidants lack this explanation. Although this review is comprehensive, only one figure or table has the disadvantage of not being very visual in its content. Personally, I would have liked to see a table showing the antioxidants considered promising and their main sources.

The author's assertion in Line95-96 is lacking in evidence. With the statement that the analysis is done in the group of 45 years and older, you are not analyzing the elderly and non-elderly separately. If there is a situation where lignan works well in the elderly, would there be no significant difference? I did not understand why the authors could say that it works well for the non-elderly. How do they differentiate between the elderly and non-elderly, which the authors did not initially define in the first place?

Is lutein, which is the subject of Line Line137-9 claim? Antioxidants in general? It seemed vague to me.

Although pomegranate juice has an antioxidant effect,it seems that the original author of citation 50 did not read the antioxidant effect as the primary cause of the improvement in cognitive function. Please checks the paper, and the author's ideas should be written separately.

I think chapter 2.2 is problematic. It is good to collect literature on the good results of antioxidants, but I think there is a lot of negative literature that shows that some (I can think of around vitamin E and Ginkgo biloba) do not show any AD preventive effect.

https://pubmed.ncbi.nlm.nih.gov/28319243/

 This does not seem fair as a review and risks creating a false perception that all one has to do is to take antioxidants. The  explaining both sides of the argument in this chapter would provide readers with more useful information while maintaining a fair perspective. I don't think it is enough to just state abstract things in Discussion.

Is Discussion required in the review? There is quite some duplicate wording, so I strongly recommend that you summarize it chapter by chapter. This simply makes it longer and is quite dreadful.

There may be a point of view to consider the effect on nerves via the gut, which is described in the conclusion, and conversely, the oxidants (pesticides?) that comes from food, etc.

Figure 1 shows that antioxidants interfere with healthy neuronal function. It is better to distinguish between ROS and antioxidant signals based on the color of the arrows. The font color in the dark areas is difficult to see (especially purple) and should be considered.

Author Response

Comment 1:

I felt that the authors and others collected a little too much good evaluation literature, but they are well studied and could have a positive impact on this field. However, I would like to point out that there are some problems with the current situation.

I think it is well researched and informative, but the duplicate sections are dreadful. I suspect that in many cases you do not add discussion sessions to your review. At least avoid duplication carefully.

It is good to point out the benefits of antioxidants, but it would be better to include the negative information as well.

More figures could be added.

There is not much visual clarity about this review, which is a drawback.

Response 1:

We are grateful for the positive feedback and the constructive insights, which helped us improve our manuscript.

Comment 2:

 Regarding the mechanism of oxidative stress generation,in the case of neurodegenerative diseases,protein aggregates and inflammatory mechanisms are involved,which may differ from simple aging. They should probably touch this point a little more. Could this argument support the difference between aging and dementia as described by the author?

Response 2:

We thank the reviewer for this suggestion. We added new sentences in the introduction:

“In the case of neurodegenerative disorders, however, additional detrimental mechanisms come into play, such as the buildup of protein aggregates [8]. This process also fosters neuroinflammation, which is not only a byproduct of neurodegeneration but also plays a crucial role in its development as it is present even before protein aggregation [9]. Additionally, the degeneration induced by protein aggregation engenders a detrimental cycle between the exhaustion of phagocytosis and the stimulation of oxidative genes, leading to overproduction of toxic oxidative species [10].”

Comment 3:

  It is interesting,but further explanation of the details of each antioxidant would be useful. What foods contain them, etc., would be of interest. However, several antioxidants lack this explanation. Although this review is comprehensive, only one figure or table has the disadvantage of not being very visual in its content. Personally, I would have liked to see a table showing the antioxidants considered promising and their main sources.

Response 3:

Thanks for suggesting this addition, we created a table that outlines the promising antioxidants and their corresponding food sources.

Comment 4:

 The author's assertion in Line95-96 is lacking in evidence. With the statement that the analysis is done in the group of 45 years and older, you are not analyzing the elderly and non-elderly separately. If there is a situation where lignan works well in the elderly, would there be no significant difference? I did not understand why the authors could say that it works well for the non-elderly. How do they differentiate between the elderly and non-elderly, which the authors did not initially define in the first place?

Response 4:

We agree with this suggestion, and updated our statement accordingly as the analyses in the original study did not split the sample based on age:

“In any case, the results suggest that lignans may be beneficial as a protective factor against cognitive decline.”

Comment 5:

 Is lutein, which is the subject of Line Line137-9 claim? Antioxidants in general? It seemed vague to me.

Response 5:

Thanks for asking for a needed clarification, we updated our sentence:

“Beside the investigations outlined above, which brought forward correlational evidence, even randomized, placebo-controlled studies reveal promising results, corroborating these findings and providing causal evidence by directly manipulating dietary regimes or by providing daily doses of supplements and evaluating their impact on cognitive functioning.”

Comment 6:

Although pomegranate juice has an antioxidant effect,it seems that the original author of citation 50 did not read the antioxidant effect as the primary cause of the improvement in cognitive function. Please checks the paper, and the author's ideas should be written separately.

Response 6:

We thank the Reviewer for their suggestion. We updated our remark to reflect the original findings:

“Thus, the authors suggest that pomegranate juice may increase task-related brain activations and consequently improve memory abilities in older adults.”

Comment 7:

 I think chapter 2.2 is problematic. It is good to collect literature on the good results of antioxidants, but I think there is a lot of negative literature that shows that some (I can think of around vitamin E and Ginkgo biloba) do not show any AD preventive effect.

https://pubmed.ncbi.nlm.nih.gov/28319243/

 This does not seem fair as a review and risks creating a false perception that all one has to do is to take antioxidants. The  explaining both sides of the argument in this chapter would provide readers with more useful information while maintaining a fair perspective. I don't think it is enough to just state abstract things in Discussion.

Response 7:

Thank you for pointing out this issue. We added a paragraph to report findings that suggest no beneficial effects of antioxidants on cognition in human studies:

“While the abovementioned investigations provide promising evidence concerning the role of antioxidants as preventive measures to contrast cognitive decline as well as slowing down neurodegeneration, other studies highlighted that antioxidant supplementation did not yield positive effects.

Broman-Fulks and colleagues [99] examined the effects of quercetin supplementation on cognitive performance in healthy adults of various age ranges. Participants were randomized to receive either quercetin or a placebo for 12 weeks. Cognitive performance was assessed using a battery of neuropsychological tests named Central Nervous System Vital Signs [100], which includes measures of memory, executive function, and processing speed. The results indicated no significant improvements in cognitive performance attributable to quercetin supplementation compared to placebo, regardless of age group. Additionally, some evidence supports the idea that resveratrol may not have positive effects on cognition. As an example, Kennedy and colleagues [101] and Wightman and colleagues [102] assessed the effects of this antioxidant on cognitive performance and cerebral blood flow. Specifically, they assessed cognitive performance with the Rapid Visual Information Processing task [103] and the Serial Subtraction task [104] while undergoing near-infrared spectroscopy (NIRS) – a noninvasive brain imaging technique – to evaluate hemodynamic responses. The results of both studies revealed that, while resveratrol or its combination with piperine – a compound with antioxidant properties found in black pepper [105] – had a dose-dependent effect on cerebral blood flow, this was not reflected in an improvement in cognitive performance.

Some important, additional insights emerge from studies specifically focused on dementia. A study conducted by Kryscio and colleagues [106] suggested that administration of vitamin E, selenium, or a combination of the two, did not affect the chances of developing dementia compared to a placebo. Furthermore, the supplementation of vitamin E in AD patients results in either beneficial or detrimental effects on cognition based on its activity. When vitamin E fosters an antioxidant effect – as measured with oxidized glutathione levels in the blood – cognitive test scores remain stable compared to a placebo condition. However, in patients that did not show oxidative stress prevention, test scores were lower than patients in the placebo condition [107]. While the results of this study support the idea that an antioxidant action can be beneficial to prevent cognitive decline, they warn against the risks of indiscriminate use of supplements, which can lead to opposite outcomes.”

Comment 8:

 Is Discussion required in the review? There is quite some duplicate wording, so I strongly recommend that you summarize it chapter by chapter. This simply makes it longer and is quite dreadful.

Response 8:

We thank the Reviewer for suggesting this change. We condensed the summarization of the chapters at the end of them in order to streamline the flow of information.

Comment 9:

 There may be a point of view to consider the effect on nerves via the gut, which is described in the conclusion, and conversely, the oxidants (pesticides?) that comes from food, etc.

Response 9:

Thanks for suggesting the addition of this information. We included a mention to pesticides in the introduction:

“Furthermore, oxidative stress may be increased by exogenous sources, for example pesticides [11,12], which are known to be one of the causes of the pathogenesis of neurodegenerative disorders [13,14].”

And a separate paragraph to describe the gut-brain axis:

“The gut-brain axis (GBA) is a complex, bidirectional communication system involving the central nervous system, the enteric nervous system, and the gut microbiota. This axis regulates key physiological processes, including food intake, immune function, and mood. The GBA allows gut-derived molecules and microbe-derived neuroactive compounds to influence brain function, and vice versa. The vagus nerve, a crucial component of the autonomic nervous system, acts as a neurometabolic sensor, detecting gut microbiota metabolites and transmitting this information to the brain [30].

Converging and recent evidence is suggestive of the central role of the GBA as a crucial substrate involved in the process of neurodegeneration [31]. Specifically, gut microbiota composition perturbations lead to increased oxidative stress, neuronal death, and neuroinflammation, culminating in the development of disorders such as Alzheimer’s disease [32,33] and Parkinson disease [34,35]. Disturbances in gut microbiota—characterized by a reduction in beneficial species and an increase in pathogenic, pro-inflammatory bacteria—lead to the production of toxic metabolites and pro-inflammatory cytokines. These changes compromise the integrity of the gut epithelial barrier, activate local and distant immune responses, and cause dysregulation of enteric neurons and glia. Subsequent dysfunctions in the blood-brain barrier emerge, triggering neuroinflammatory responses and predisposing neurons and glial cells to apoptosis, particularly in brain regions such as the hippocampus and cerebral cortex, contributing to the development of neurodegenerative disorders [36].

Consumptions of antioxidants fosters neuroprotective effects and can modulate gut microbiota composition, increasing beneficial bacteria and reducing pathogenic species, also improving gut barrier integrity and blood-brain barrier permeability [31].”

Comment 10:

 Figure 1 shows that antioxidants interfere with healthy neuronal function. It is better to distinguish between ROS and antioxidant signals based on the color of the arrows. The font color in the dark areas is difficult to see (especially purple) and should be considered.

Response 10:

We thank the Reviewer for these suggestions, we updated the figure accordingly.

Reviewer 3 Report

In this review, the authors aim at collecting evidence that focused on the role of a variety of antioxidants and antioxidant-rich foods in improving or stabilizing cognitive functions, memory, and Alzheimer's disease, the most common neurodegenerative disorder. It is readable and interesting.

There are several comments on this manuscript.

1.     It would be more readable and attractive if the authors detailed display the common antioxidants and antioxidant-rich foods, as well as their mechanisms, in neuroprotection in a table or figure.

2.     Aging is a process affecting the biological changes of whole body. According to the content of this manuscript, it would be more accurate to substitute the word “aging” for “cognitive decline” in the title.

3.     There are also literatures that reported the detrimental or negative effects of excessive intake of antioxidants or vitamins, which should be mentioned or discussed in the manuscript.

4.     What’s “3MS score” in line 82?  What’s the full name of MS?

Author Response

Comment 1:

In this review, the authors aim at collecting evidence that focused on the role of a variety of antioxidants and antioxidant-rich foods in improving or stabilizing cognitive functions, memory, and Alzheimer's disease, the most common neurodegenerative disorder. It is readable and interesting.

Response 1:

We thank the Reviewer for their positive feedback and their suggestions.

Comment 2:

There are several comments on this manuscript.

It would be more readable and attractive if the authors detailed display the common antioxidants and antioxidant-rich foods, as well as their mechanisms, in neuroprotection in a table or figure.

Response 2:

Thanks for this recommendation. We added a table that details the different studies that have been analyzed in the manuscript, including the antioxidants investigated and the main results for each.

Comment 3:

Aging is a process affecting the biological changes of whole body. According to the content of this manuscript, it would be more accurate to substitute the word “aging” for “cognitive decline” in the title.

Response 3:

Thank you for the suggestion, we updated the title accordingly.

Comment 4:

There are also literatures that reported the detrimental or negative effects of excessive intake of antioxidants or vitamins, which should be mentioned or discussed in the manuscript.

Response 4:

We thank the Reviewer for this comment. Accordingly, we added a chapter that presents contrasting evidence in this field:

“While the abovementioned investigations provide promising evidence concerning the role of antioxidants as preventive measures to contrast cognitive decline as well as slowing down neurodegeneration, other studies highlighted that antioxidant supplementation did not yield positive effects.

Broman-Fulks and colleagues [99] examined the effects of quercetin supplementation on cognitive performance in healthy adults of various age ranges. Participants were randomized to receive either quercetin or a placebo for 12 weeks. Cognitive performance was assessed using a battery of neuropsychological tests named Central Nervous System Vital Signs [100], which includes measures of memory, executive function, and processing speed. The results indicated no significant improvements in cognitive performance attributable to quercetin supplementation compared to placebo, regardless of age group. Additionally, some evidence supports the idea that resveratrol may not have positive effects on cognition. As an example, Kennedy and colleagues [101] and Wightman and colleagues [102] assessed the effects of this antioxidant on cognitive performance and cerebral blood flow. Specifically, they assessed cognitive performance with the Rapid Visual Information Processing task [103] and the Serial Subtraction task [104] while undergoing near-infrared spectroscopy (NIRS) – a noninvasive brain imaging technique – to evaluate hemodynamic responses. The results of both studies revealed that, while resveratrol or its combination with piperine – a compound with antioxidant properties found in black pepper [105] – had a dose-dependent effect on cerebral blood flow, this was not reflected in an improvement in cognitive performance.

Some important, additional insights emerge from studies specifically focused on dementia. A study conducted by Kryscio and colleagues [106] suggested that administration of vitamin E, selenium, or a combination of the two, did not affect the chances of developing dementia compared to a placebo. Furthermore, the supplementation of vitamin E in AD patients results in either beneficial or detrimental effects on cognition based on its activity. When vitamin E fosters an antioxidant effect – as measured with oxidized glutathione levels in the blood – cognitive test scores remain stable compared to a placebo condition. However, in patients that did not show oxidative stress prevention, test scores were lower than patients in the placebo condition [107]. While the results of this study support the idea that an antioxidant action can be beneficial to prevent cognitive decline, they warn against the risks of indiscriminate use of supplements, which can lead to opposite outcomes.”

Comment 5:

What’s “3MS score” in line 82?  What’s the full name of MS

Response 5:

This acronym stands for Mini-Mental State Examination, we included the full name at first occurrence.

Round 2

Reviewer 1 Report

Authors have considered all this referee's suggestions and have much improved the manuscript.

From my point of view the manuscript, in this new version, is suitable for pubblication.

Authors have considered all this referee's suggestions and have much improved the manuscript.

From my point of view the manuscript, in this new version, is suitable for pubblication.

Reviewer 2 Report

A few tables have been added to help organize the information.

Also, both sides of the issue have been addressed, and I believe the review is fairly presented.

I think this is a review that will have a positive impact on the field.

I don't think it is the authors' fault, but Table 1 is divided by page, so it would be better to unify the pages on either page.

No corrections are proposed other than Table 1. See comments above regarding Table 1.